# Colloidal Silver Hydrogen Peroxide: New Generation Molecule for Management of Phytopathogens

Hosapura Shekhararaju Mahesha [1,2,*], Jayasuvarnapura Umapathi Vinay [1],
Medikeripura Rekhyanaik Ravikumar [1], Suryanarayana Visweswarashastry [1],
Manikyanahalli Chandrashekhara Keerthi [2], Hanamant Mudakappa Halli [2], Shadi Shokralla [3,4],
Tarek K. Zin El-Abedin [5], Eman A. Mahmoud [6] and Hosam O. Elansary [7,*]

1    Department of Plant Pathology, University of Agricultural Sciences (UAS), Dharwad 580 005, India;
     juvinay@gmail.com (J.U.V.); ravikumarmr@uasd.in (M.R.R.); suryanarayanav@uasd.in (S.V.)
2    ICAR—Indian Grassland and Fodder Research Institute, Jhansi 284 003, India;
     keerthi.mc@icar.gov.in (M.C.K.); hanamant.halli@icar.gov.in (H.M.H.)
3    Centre for Biodiversity Genomics, University of Guelph, Guelph, ON N1G 2W1, Canada;
     sshokral@uoguelph.ca
4    Department of Integrative Biology, University of Guelph, Guelph, ON N1G 2W1, Canada
5    Department of Agriculture & Biosystems Engineering, Faculty of Agriculture (El-Shatby),
     Alexandria University, Alexandria 21545, Egypt; drtkz60@gmail.com
6    Department of Food Industries, Faculty of Agriculture, Damietta University, Damietta 34511, Egypt;
     emanmail2005@yahoo.com
7    Plant Production Department, College of Food & Agriculture Sciences, King Saud University,
     Riyadh 11451, Saudi Arabia
*    Correspondence: hsmaheshhunsur@gmail.com or mahesha.hs@icar.gov.in (H.S.M.);
     helansary@ksu.edu.sa (H.O.E.)

**Abstract:** Plant pathogenic fungi and bacteria are a significant threat to global commercial crop production resulting in increased cost of production, reduced crop establishment and productivity. An effort was made to study the antimicrobial activity of silver hydrogen peroxide (SHP) against selected plant pathogenic fungi and bacteria under in vitro conditions. Higher antibacterial activity of SHP was observed against *Xanthomonas axonopodis* pv. *citri* (*Xac*; 39.67 mm), *Xanthomonas citri* pv. *punicae* (*Xap*; 39.00 mm), and *Ralstonia solanacearum* (*Rs*; 36.67 mm) at 500 ppm concentration. SHP was superior to streptocycline (500 ppm) against *Xac* (25.33 mm) and *Xcp* (22.67 mm) at 100 ppm. The soil-borne fungi viz., *Pythium aphanidermatum* and *Fusarium solani* failed to initiate mycelium growth on PDA at the concentration of 5000 ppm and above. The average size of SHP particles was 462 nm in diameter, and 73.40% of particles had the size of 378 nm, which reflects the particles present in SHP solution in the form of colloids. The effective doses (100–5000 ppm) did not show any phytotoxicity symptoms in plants, while leaf necrosis was noticed at 10,000 ppm after four days of application. SHP (≤5000 ppm) can be used to effectively manage both fungal and bacterial plant pathogens by a single application. Further field studies need to be conducted for validation and commercial use of SHP.

**Keywords:** silver hydrogen peroxide; soil-borne pathogens; colloid; phytotoxicity; antibacterial; streptocycline

## 1. Introduction

Agricultural productivity is threatened by biotic and abiotic stresses resulting in loss of productivity in terms of both quality and quantity [1]. Biotic stresses such as pathogens, insects, and weeds account for 20 to 40 per cent of global crop loss every year [2]. Both pests and diseases cause direct and indirect losses to the crop. If we assume an average crop loss of 20 per cent, with the present gross value of our agriculture produce at Rs 7 lakh crore, the average loss comes to Rs. 140,000 crores [3]. Among biotic stresses, soil-borne pathogens such as fungi,

bacteria, and nematodes are a major threat to global commercial crop production and result in reduced crop establishment, increased cost of production and reduced productivity [4]. Plant pathogens such as *Pythium* spp., *Fusarium* spp., *Sclerotium rolfsii*, *Rhizoctonia* spp., and *Ralstonia solanacearum* have a strong survival in the soil as dormant/resting structures in the absence of their host plant and have the capacity to cause disease [5].

Moreover, accurate diagnosis of disease caused by different pathogens is challenging because most of the disease symptoms are similar to those caused by abiotic stresses, making the disease harder to manage [6]. Inadequate disease diagnosis often results in the application of unsuitable agrochemicals, which further aggravates the problem and makes farming unprofitable. To manage these diseases, farmers rely on fungicides, antibiotics, and fumigants at regular intervals. However, many presently available chemical pesticides are highly toxic and nonbiodegradable, persist in food products, and causes environmental pollution.

Hydrogen peroxide ($H_2O_2$) is a versatile chemical compound with good germicidal activity (bactericidal, virucidal, sporicidal, and fungicidal properties). $H_2O_2$ has a widespread antibacterial and antifungal activity; therefore, it is used as a surface disinfectant in food, medical, water treatment, as well as industrial fields [7,8]. Hydrogen peroxide inactivates *E. coli* at 30–100 ppm via DNA damage [9]. The main drawback of $H_2O_2$ is its strong oxidising nature, which decomposes it into water and oxygen, thereby reducing its efficiency.

The use of metal ions as antimicrobial agents has become the best alternative to synthetic pesticides, and technological advancements have become widespread, making their production more economical [10]. Silver is a very attractive and versatile metal with several applications due to its unique and remarkable properties, such as enhanced permeability, retention effect, and antimicrobial activity. Silver is more toxic to microorganisms than other metals and shows reduced toxicity for mammalian cells. Adding silver to $H_2O_2$ further increases its antimicrobial activity as it plays the role of a stabiliser and an activator [11]. Strong antibacterial activities of silver ions and $H_2O_2$ have been reported against three different genera of waterborne bacterial pathogens. However, limited information is available on the application of hydrogen peroxide and silver ions in the agriculture ecosystem, especially in plant disease management. Therefore, this study was undertaken to examine the efficacy of commercially available silver hydrogen peroxide (SHP) against major plant pathogenic fungi and bacteria under in vitro conditions and to evaluate their toxicity on crop plants.

## 2. Material and Methods

### 2.1. Chemical Composition and Source

A commercial formulation of silver hydrogen peroxide, i.e., Alstasan Silvox® (Manufacturer: Chemtex Speciality Ltd. Kolkata, West Bengal, India), was evaluated for antimicrobial activity against plant pathogens. It consists of silver ions (500 ppm) and hydrogen peroxide (49.5–50% Min. 48% *v/v*) with a pH of 1.5–2.0.

### 2.2. Pathogen Isolation

Pathogens used in the study, their host plant, and collection place were presented in Table 1.

The fungal pathogens such as *P. aphanidermatum* and *F. solani* were isolated following the standard tissue isolation technique. Infected ginger rhizomes and pseudostem were washed in a gentle stream of running water, cut into 5 mm pieces and surface sterilised with NaOCl solution (1%) for one min followed by three quick serial washes with sterile distilled water. Then, they were placed on PDA amended with selective antibiotics for *P. aphanidermatum* [12], and only PDA was used for *F. solani*. Fungal purification and pathogenicity test were conducted following standard methods [13].

**Table 1.** List of plant pathogens used for the study.

| Sl. No. | Pathogen | Host | Plant Part Used for Isolation | Latitude and Longitudes |
|---|---|---|---|---|
| 1 | *Pythium aphanidermatum* | Ginger- *Zingiber officinale* | Pseudostem | N14.5296 E75.0210 |
| 2 | *Fusarium solani* | Ginger- *Z. officinale* | Rhizomes | N14.5296 E75.0210 |
| 3 | *Ralstonia solanacearum* | Tomato- *Solanum lycopersicum* | Stem | N15.2930 E74.5909 |
| 4 | *Xanthomonas citri* pv. *punicae* | Pomegranate- *Punica granatum* | Fruits | N16.7641 E75.7478 |
| 5 | *X. axonopodis* pv. *citri* | Lemon- *Citrus* sp. | Fruits | N15.2930 E74.5909 |

Fruits showing characteristic symptoms of pomegranate bacterial blight and citrus canker were collected from the infected garden. Infected samples were cut into 5 mm size pieces, followed by surface sterilisation with sodium hypochlorite (1%) for 1 min and washed with sterile double distilled water thrice to remove traces of sodium hypochlorite. Then the bits were suspended in a test tube containing 10 mL sterile distilled water and allowed to ooze out bacterial cells into water blank for 20 min at room temperature. The bacterial pathogens were isolated by adopting the serial dilution technique on nutrient glucose agar ($10^5$–$10^6$) by the spread plate method [14].

Bacterial wilt-infected tomato stems were collected and washed in running tap water. Infected stem samples were cut into small pieces (5 mm) and surface sterilised, as mentioned above. *Ralstonia solanacearum* was isolated following the serial dilution technique ($10^5$–$10^6$) on Casamino acid-Peptone-Glucose (CPG) agar medium [15,16]. Pure cultures of bacterial pathogens were stored in 20% sterile glycerol at $-80\ ^{\circ}$C until further use.

### 2.3. Morphological and Biochemical Characterisation of Test Pathogens

Five mm discs of *P. aphanidermatum* and *F. solani* isolate were placed at the centre of the Petri plates (90 mm dia.) having sterile PDA, and plates were incubated at $26 \pm 1\ ^{\circ}$C for eight days. Cultural and microscopic observations were taken on parameters viz., colony colour, mycelium colour, presence or absence of septa, presence or absence of spore, and size of spores by using Leica DM 2500 LED microscope (400× magnification). The biochemical characteristics of bacterium such as Gram's staining, potassium hydroxide solubility test (KOH test), starch hydrolysis, hydrogen sulphide production, and catalase production tests were conducted as per the standard procedures described by [17,18].

### 2.4. DNA Isolation, PCR Amplification, and Sequencing

Total genomic DNA of fungal (*P. aphanidermatum* and *F. solani*) and bacterial (*X. citri* pv. *Punicae*, *X. axonopodis* pv. *Citri* and *R. solanacearum*) pathogens were extracted using HiPurA™ fungal DNA purification kit for fungi and HiPurA™ bacterial genomic DNA purification kit (Himedia Pvt. Ltd. Mumbai, India) respectively, following manufacturer's instruction. PCR amplification of internal transcribed spacer (ITS) region of fungi was performed using universal fungal primer, i.e., ITS1 and ITS4 [19], and amplification of 16S rRNA gene of bacterial genomic DNA was conducted using universal primer, 27F (5′-AGAGTTTGATCCTGGCTCAG-3′) and 1492R (5′-GGTTACCTTGTTACGACTT-3′) for confirmation of pathogens at DNA level [20]. Amplified products were separated on 1.2% agarose gel using an electrophoresis unit. Amplified PCR products were purified and sequenced (SciGenome labs, Kerala, India) and resulting gene sequences were aligned using BioEdit software. Sequences of targeted regions of study pathogens were compared with earlier deposited reliable sequences to find sequence similarities using nBLAST, NCBI (http://www.ncbi.nlm.nih.gov/blast/) accessed on 23 August 2021.

## 2.5. Antibacterial Assay

The antibacterial activity of silver hydrogen peroxide was evaluated against *X. citri* pv. *Punicae* and *X. axonopodis* pv. *Citri* by paper disk assay [21] and *R. solanacearum* by following the agar well diffusion method. Loopful of each 48-h-old bacterial culture was inoculated into nutrient broth separately and incubated at $30 \pm 1$ °C in a shaking incubator with a speed of 150 rpm. The bacterial cells were harvested 36 h after incubation by centrifugation at 6000 rpm for 5 min and washed twice with deionised water, and resuspended in deionised water. The bacterial suspensions were adjusted to a concentration of $10^6$–$10^7$ cfu/mL. Later, 50 µL bacterial suspension was evenly spread on solidified nutrient glucose agar. The sterilised filter paper discs (Whatman No.1; 5 mm diameter) were soaked in different concentrations of silver hydrogen peroxides (10, 25, 50, 100, 250, and 500 ppm) for 1 h, and impregnated discs were placed on the culture plate of *X. citri* pv. *Punicae* and *X. axonopodis* pv. *Citri*. For *R. solanacearum*, wells (5 mm dia.) were made on bacterial cells spread nutrient glucose agar. Later, different concentrations of silver hydrogen peroxide were added to wells. Silver nitrate (3 mM) and streptocycline (500 ppm) were taken as a positive control, and sterile distilled water served as a negative control. Inoculated plates were incubated at $28 \pm 1$ °C for 48 h. Each treatment was replicated thrice, and the zone of inhibition was recorded by measuring diameter (mm).

## 2.6. Antifungal Assay

Different concentrations of silver hydrogen peroxide (1000, 5000, 10,000, 20,000, 30,000, and 40,000 ppm) were tested to determine the antifungal properties against soilborne fungal pathogens using the poison food technique. Required quantity of silver hydrogen peroxide (to prepare different concentrations) was added to 100 mL sterile, molten potato dextrose agar, thoroughly mixed, and aseptically poured into disposable presterilised Petri dishes. After solidification, 5-d-old actively growing 5 mm mycelial disc of *P. aphanidermatum* and *F. solani* was placed on the centre of the plate separately for each concentration, and the inoculated plates were incubated at $25 \pm 1$ °C and $28 \pm 1$ °C, respectively. Metalaxyl 4%+ mancozeb 64% WG at 1500 ppm and carbendazim 50% WP at 1000 ppm concentration was used as a positive control for *P. aphanidermatum* and *F. solani,* respectively. Negative control was maintained without the addition of fungicides. Each set of experiments was replicated thrice. The per cent mycelial inhibition over control was calculated using the formula given [22].

## 2.7. Characterisation of Silver Hydrogen Peroxide Particles

The particle size analyser (Nicomp, Orlando, FL, USA) was used to characterise Alstasan Silvox®, for which 1 mL of the commercial formulation was suspended in 10 mL of sterile distilled water and sonicated for 10 min for uniform distribution. Further, 3 mL of sonicated solution was subjected to particle size analysis.

## 2.8. Phytotoxicity on Cotton Plants

Phytotoxicity studies were conducted under glasshouse conditions to determine the safe and effective doses of silver hydrogen peroxide. Thirty-day-old cotton plants (n = 3) were used for each concentration and in each plant, three leaves were spotted with different concentrations of SHP solution (10, 25, 50, 100, 250, 500, 1000, 5000, and 10,000 ppm). The observations were recorded on 1, 3, 5, and 7 days after spotting for expression of various symptoms of phytotoxicity (hyponasty, epinasty, vein clearing, stunting, scorching, necrosis, and chlorosis) using the following visual phytotoxicity scale (0–10) [23].

## 2.9. Statistical Analysis

The per cent mycelial growth inhibition data of fungal pathogen was checked for normality and homogeneity and then transformed using the arcsine transformation. Subsequently, the data were analysed using a completely randomised design. The inhibition

zone was recorded in millimetres, and both data were analysed using one way ANOVA in SAS version 9.3. The significance level was judged at *p* value < 0.01.

## 3. Results

### 3.1. Morphological and Biochemical Characterisation of Study Pathogens

The colony of *P. aphanidermatum* on PDA was smooth with white, cottony aerial filamentous mycelium, later turning to greyish-white colour. Microscopic results show mycelium hyaline, coenocytic, and antheridium were club-shaped and intercalary, whereas oogonium terminal, smooth, globose and oospores were golden-yellow with size ranges from 20–22 μm in diameter.

On potato dextrose agar, colonies of *F. solani* were thick, cottony white with floccose mycelium with a medium growth rate of 8 mm/day. Under the microscope, mycelium was hyaline and septate. The microconidia were oval to cylindrical, hyaline, ranging from 5.5 to 12 × 2 to 5.4 μm (n = 25), while macroconidia were hyaline, slightly curved with 3 to 5 septa and measured about 3.5–6.5 × 13.45–32.74 μm (n = 25). Chlamydospores were thickly walled, both intercalary and terminal, and occurred both single and in a chain.

Colonies of *R. solanacearum* were observed 36 h after inoculation. Virulent colonies were well isolated, slimy, fluidal, irregular with pink to light red centre with dull white border on TZC medium. On Grams staining, bacterial cells were Gram-negative in reaction and morphologically rod-shaped. Bacterium showed a positive reaction to all biochemical tests viz., KOH, starch hydrolysis, hydrogen sulphide production, and catalase test. *X. citri* pv. *Punicae* colonies appeared as light yellow, raised, convex and glistening with pale yellow to dark yellow colour after 72 h of incubation. The *X. axanopodis* pv. *Citri* colonies were creamy yellow with copious slime, raised and convex in appearance. On Grams staining, both the bacteria were Gram-negative in reaction and showed positive reaction to all biochemical tests (Table 2).

**Table 2.** Biochemical characterisation of the bacterial pathogens.

| Biochemical Test | *X. axonopodis* pv. *Citri* | *X. citri* pv. *Punicae* | *R. solanacearum* |
|---|---|---|---|
| Gram staining | − | − | − |
| KOH test | + | + | + |
| Starch hydrolysis | + | + | + |
| Hydrogen sulphide production | + | + | + |
| Catalase test | + | + | + |

Note: + indicates the positive reaction and—indicate the negative reaction to biochemical tests.

### 3.2. Molecular Identification

Molecular identification of pathogens was based on DNA sequence analysis, which helped with species confirmation. The amplified PCR product of *P. aphanidermatum* and *F. solani* yielded an expected amplicon size of 800 and 550 bp for the universal fungal primer (ITS1 & ITS4), respectively. Both pathogens were sequenced at SciGenom Labs, Kerala. The sequences were aligned using Bioedit 10 software and confirmed through the NCBI database with the help of nucleotide BLAST. Based on the nucleotide sequences comparison, pathogens were confirmed as *P. aphanidermatum* and *F. solani* with 98 and 99 per cent similarity with already deposited sequences, respectively. The sequences of *P. aphanidermatum* and *F. solani* were deposited in the NCBI GenBank database under the accession number MT012240 and MN853411, respectively.

Similarly, PCR amplification was performed for bacterial pathogens, *X. citri* pv. *Punicae*, *X. axonopodis* pv. *Citri* and *R. solanacearum* by using 16S rRNA gene primer. The results revealed that all three bacteria produced an amplicon size of 1.4 kb. The sequences were aligned, and the BLAST search at the NCBI database confirmed the bacterial pathogenic species at the molecular level. The analysed results showed that *X. axonopodis* pv. *Citri*, *X. citri* pv. *Punicae* (MN252384) and *R. solanacearum* (MN853412) were 98–99 per cent similar to other isolates in NCBI.

### 3.3. Antibacterial Activity

Different concentrations (10–500 ppm) of SHP were evaluated against three bacterial plant pathogens, viz., *X. citri* pv. *Punicae*, *X. axonopodis* pv. *Citri* and *R. solanacearum*. SHP recorded a maximum inhibition zone of 39.67, 39.00, and 36.67 mm (diameter) against *Xac*, *Xcp*, and *Rs* at 500 ppm concentration, respectively (Figure 1). SHP (100 ppm) was on par with streptocycline (500 ppm) with respect to antibacterial activity against *Xac* (25.33 mm) and *Xcp* (22.67 mm) (Figures 2 and 3). SHP showed a 19.00 mm inhibition zone against *Rs* at 50 ppm concentration which was on par with streptocycline (18.33 mm) at 500 ppm (Figure 4). Bulk silver nitrate showed 13.00, 13.00, and 13.33 mm zone of inhibition against *Xac*, *Xcp*, and *Rs*, respectively.

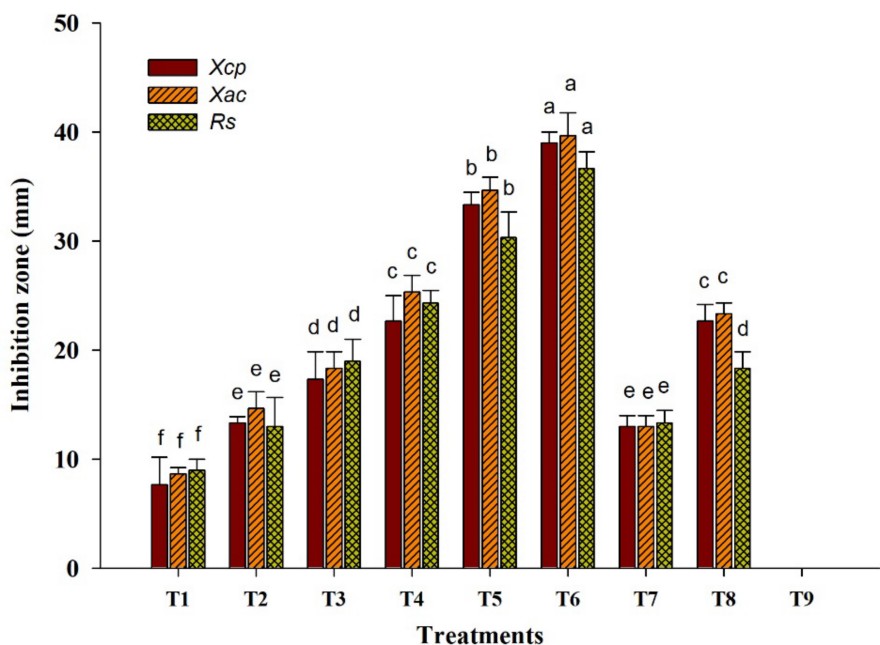

Whereas, $T_1$ : Silver hydrogen peroxide (SHP) @ 10 ppm, $T_2$ : SHP @ 25 ppm, $T_3$ : SHP @ 50 ppm, $T_4$ : SHP @ 100 ppm, $T_5$ : SHP @ 250 ppm, $T_6$ : SHP @ 500 ppm, $T_7$ : Silver nitrate @ 3 mM, $T_8$ : Streptocycline @ 500 ppm, $T_9$ : Water control

**Figure 1.** Bio-efficacy of different concentrations of silver hydrogen peroxide against bacterial plant pathogens (*X. citri* pv. *Punicae*, *X. axonopodis* pv. *citri*, and *R. solanacearum*) under in vitro condition. In a bar having a similar letter(s) do not differ significantly at 5% level, whereas bars with the dissimilar letter(s) differ significantly at the same level (DMRT).

### 3.4. Antifungal Activity

Bioefficacy of SHP was tested in vitro against *P. aphanidermatum* and *F. solani* under condition. The results revealed that SHP failed to inhibit both test pathogens at 1000 ppm. However, 100% mycelial growth inhibition was recorded at 5000 ppm and above concentration against both fungal pathogens. Positive chemical check with carbendazim (1000 ppm) and metalaxyl + mancozeb (1500 ppm) also showed complete inhibition of *F. solani* and *P. aphanidermatum* (Figure 5), respectively. Bulk silver nitrate failed to show antifungal activity against both pathogens (Table 3).

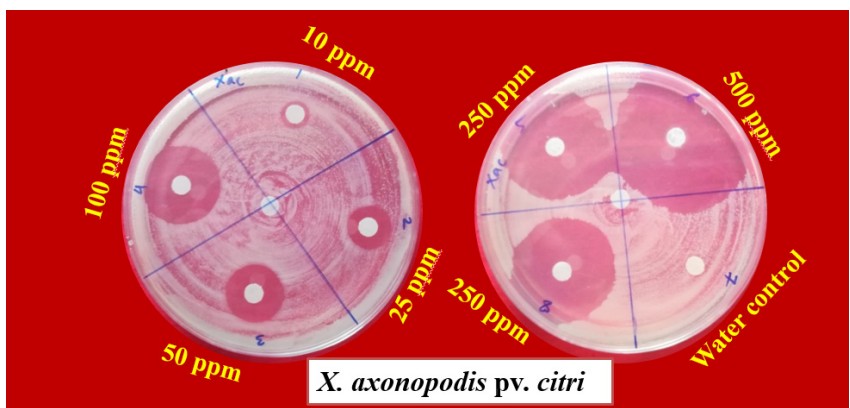

**Figure 2.** Bactericidal activity of silver hydrogen peroxide against *X. axonopodis* pv. *citri* in paper disc method under in vitro condition.

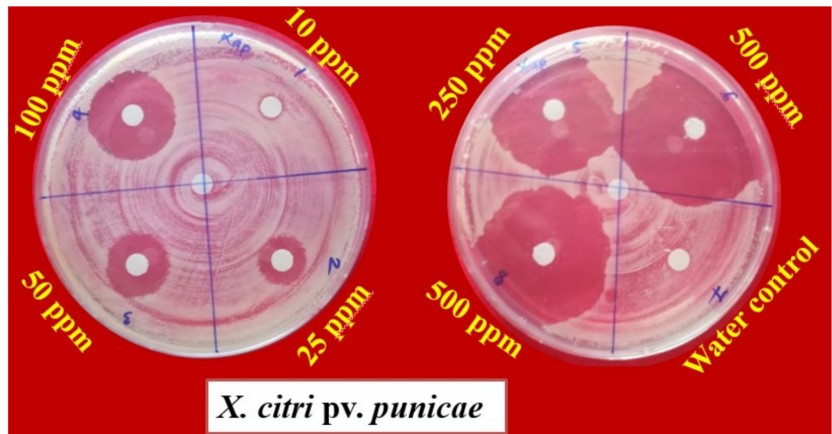

**Figure 3.** Antibacterial activity of colloidal silver hydrogen peroxide against *X. citri* pv. *punicae* in paper disc method under in vitro condition.

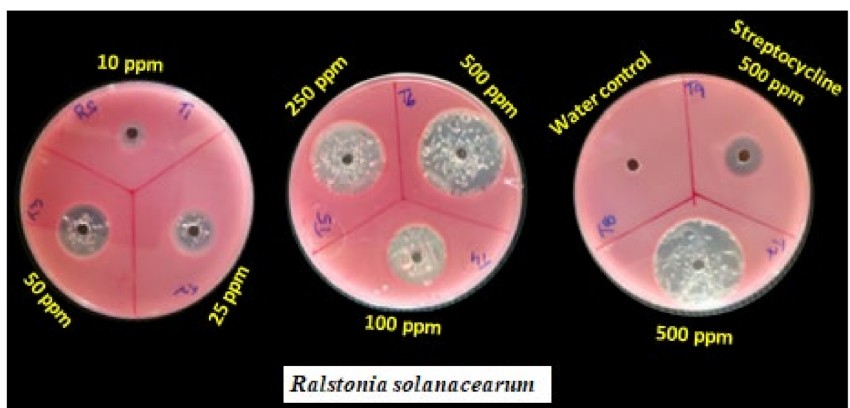

**Figure 4.** Efficacy of different concentrations of silver hydrogen peroxide (10–500 ppm) against *R. solanacearum* in agar well diffusion method.

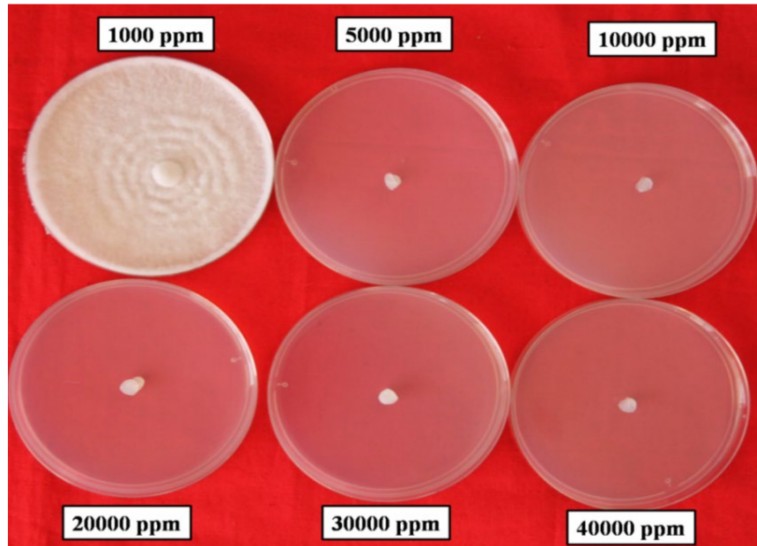

**Figure 5.** Antifungal activity of various concentrations of silver hydrogen peroxide (10,000–40,000 ppm) against *P. aphanidermatum* in poison food method under in vitro condition.

**Table 3.** Efficacy of silver hydrogen peroxide against *F. solani* and *P. aphanidermatum* in poison food method under in vitro condition.

| Treatments | Concentrations (ppm) | Mycelial Inhibition (%) | |
|---|---|---|---|
| | | *F. solani* | *P. aphanidermatum* |
| Silver hydrogen peroxide | 1000 | 0.00 [c] (0.00) * | 0.00 [c] (0.00) |
| Silver hydrogen peroxide | 5000 | 100.00 [a] (90.00) | 100.00 [a] (90.00) |
| Silver hydrogen peroxide | 10,000 | 100.00 [a] (90.00) | 100.00 [a] (90.00) |
| Silver hydrogen peroxide | 20,000 | 100.00 [a] (90.00) | 100.00 [a] (90.00) |
| Silver hydrogen peroxide | 30,000 | 100.00 [a] (90.00) | 100.00 [a] (90.00) |
| Silver hydrogen peroxide | 40,000 | 100.00 [a] (90.00) | 100.00 [a] (90.00) |
| Silver nitrate | 3 mM | 3.70 [b] (11.09) | 3.70 [b] (11.09) |
| Carbendazim 50% WP | 1000 | 100.00 [a] (90.00) | - |
| Metalaxyl 4%+ Mancozeb 64% WG | 1500 | - | 100.00 [a] (90.00) |
| S. Em. ± | | 0.29 | 0.48 |
| CD @ 1% | | 0.88 | 1.40 |

* Arc sine value. LSD assigns same letter to different treatments having at par effects during mean comparison.

### 3.5. Analysis of Particles Size in Silver Hydrogen Peroxide

The commercial formulation of SHP (Alstasan Silvox®) was characterised using a particle size analyser. SHP particles had an average diameter of 462 nm (Intensity weight-Gaussian distribution). Nicomp intensity weight distribution analysis of SHP solution showed two different peaks, which indicates the presence of two size ranges of particles. The majority of particles (73.40%) in the SHP solution have a particle size of 378 nm, followed by 26.60% with 578 nm particle size (Figure 6). The particles in the silver hydrogen peroxide solution were present in the form of colloids.

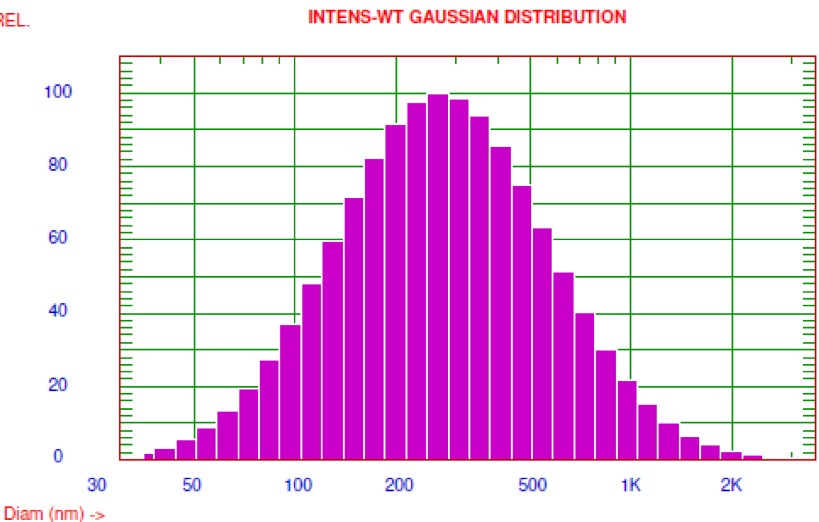

**Figure 6.** Number frequency histogram representing the particle size distribution of silver hydrogen peroxide on the linear scale using the particle size analyser.

### 3.6. Phytotoxicity Study

Phytotoxicity starts at the concentrations of 10,000 ppm and above, with symptoms such as leaf necrosis and scorching (5 rating in 0–10 rating scale) on cotton leaves at four days after application (Figure 7). However, effective doses (100–5000 ppm) of SHP were safe for the plants without any phytotoxic effect (0 rating) (Table 4).

**Table 4.** Phytotoxicity of different concentrations of colloidal silver hydrogen peroxide on cotton leaves.

| SHP Concentration (ppm) | Phytotoxicity Rating |
|---|---|
| 10 | 0 * |
| 25 | 0 |
| 50 | 0 |
| 100 | 0 |
| 250 | 0 |
| 500 | 0 |
| 1000 | 0 |
| 5000 | 0 |
| 10,000 | 5 |

* Mean of 9 replication, Phytotoxicity rating scale (0–10) was used for scoring (0 = No symptoms, 1 = Very slight discoloration, 2 = More severe, but not lasting, 3 = Moderate and more lasting, 4 = Medium and lasting, 5 = Moderately heavy, 6 = Heavy, 7 = Very heavy, 8 = Nearly destroyed, 9 = Destroyed, 10 = Completely destroyed).

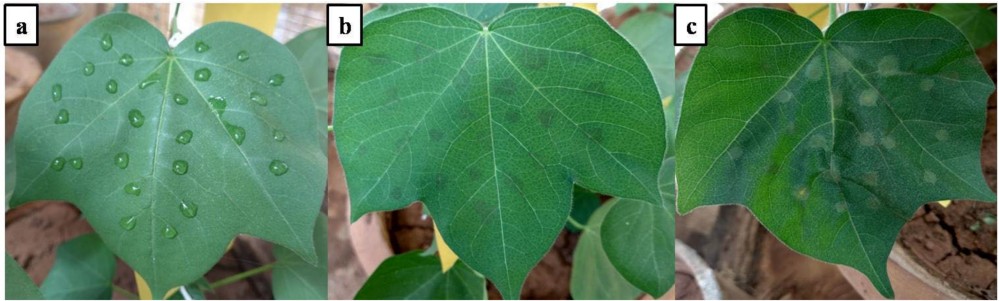

**Figure 7.** Phytotoxicity of different concentrations of colloidal silver hydrogen peroxide on cotton plants after seven days of inoculation under glasshouse condition. (**a**) Spotting on leaves; (**b**) No phytotoxicity at 5000 ppm; (**c**) Phytotoxicity at 10,000 ppm.

## 4. Discussion

In modern agriculture, the management of diseases caused by soil-borne plant pathogenic fungi and bacteria is challenging due to limited or unavailable resistance sources in germplasm, the evolution of virulent race or strains of pathogens, and the development of resistance against synthetic pesticides [24,25]. These have led to the frequent application of pesticides at regular intervals and higher doses. Consequently, the persistence of pesticides in food products is increasing at an alarming rate, becoming a global threat [26,27] further aggravated by the wrong diagnosis of diseases and erroneous application of pesticides [28].

In order to overcome these problems, it is necessary to search for eco-friendly, novel, broad-spectrum antimicrobial agents that may act against various pathogens to manage future crop protection risks. Silver hydrogen peroxide (SHP) is one such antimicrobial compound extensively used as a disinfectant in water purification [29]. The antimicrobial activity of HSP-Huwa-San peroxide® ($H_2O_2$ + Silver ions) is higher than hydrogen peroxide alone; ionic silver present in the HSP enhances the interaction of HSP with the bacterial cell surface rather than acting directly as biocide at tested concentration [11].

Morphological characterisation helps in assigning isolated pathogens to different genera. However, it was challenging to rely on only morphological characterisations to identify fungi and bacteria at the species level because a large number of species share morphological characteristics [30]. Most researchers agree on the importance of the combined use of molecular and morphological characterisation for accurately identifying pathogens [31]. Morphological and molecular characterisation reveals the associations of *P. aphanidermatum* and *F. solani* with ginger rhizome rot from different parts of India [32]. Several studies showed the association of *R. solanacearum*, *X. axonopodis* pv. *Citri*, and *X. citri* pv. *punicae* with bacterial wilt of tomato, citrus canker, and bacterial leaf blight of pomegranate, respectively [33–35].

Silver hydrogen peroxide showed good antibacterial activity against all three tested bacterial plant pathogens, i.e., *X. citri* pv. *punicae, X. axonopodis* pv. *Citri*, and *R. solanacearum*. SHP showed antibacterial activity against bacterial pathogens at 100 ppm concentration, which was on par with the antibacterial activity of streptocycline at 500 ppm. Strong antibacterial activity was noticed against *Escherichia coli*, *Proteus mirabilis*, and *Klebsiella pneumoniae* both in suspension and on the surface due to the combined use of hydrogen peroxide and silver ions [36]. Silver ion present in huwa-san peroxide (HSP: 70–500 ppm) had a limited effect on the metabolic activity of bacteria pathogens, but the antibacterial activity of HSP was due to hydrogen peroxide rather than silver [11]. The mechanism of hydrogen peroxide antimicrobial activity is due to the production of destructive hydroxyl free radicals in the Fenton reaction that causes oxidative damage to protein, membrane lipids, DNA, and other cell components [8,37]. The silver ions do not directly act as biocide; instead, they enhance the activity of HSP with the bacterial cell surface.

Multicellular organisms require a higher dose of pesticides for management compared with unicellular organisms. Soil-borne fungal pathogens require a higher concentration of fungicides compared with other pathogens due to their adaptability and complex metabolic activity. SHP failed to inhibit the tested fungal pathogens at 1000 ppm but achieved a hundred per cent inhibition of radial mycelial growth recorded at 5000 ppm. Many studies have reported the antifungal activity of silver hydrogen peroxide. In the antifungal activity of ionic hydrogen peroxide at the rate of 1 mL/m$^3$, it was found that it inactivates the fungi within five minutes of treatment [38]. Colloidal silver showed a strong antibacterial therapeutic agent against pathogenic drug-resistant bacteria, fungi, and yeast [39,40].

The majority of particles in the SHP solution has a particle size of 378 nm, followed by 26.60% with 578 nm sized particles. The silver hydrogen peroxide solution particles were present in the form of colloids, which enhanced the antibacterial activity of the silver nanoparticles due to their colloidal stability in the medium [41].

## 5. Conclusions

Broad-spectrum molecules are desirable in plant disease management due to their added benefits of single application against multiple pathogens. The silver hydrogen peroxide is one such novel, potential antimicrobial compound which effectively manages both the soil-borne (*Pythium aphanidermatum*, *Fusarium solani*, *Ralstonia solanacearum*) and foliar pathogens (*Xanthomonas* spp.) Silver hydrogen peroxide is effective in controlling these pathogens at lower concentrations. However, further studies are required to assess its effect and utility as one of the components in integrated disease management.

**Author Contributions:** Conceptualisation, H.S.M., J.U.V., M.R.R. and S.V.; design, H.S.M., J.U.V., M.R.R. and S.V.; methodology, H.S.M., J.U.V., M.R.R. and S.V.; validation, H.S.M., J.U.V., M.R.R. and S.V.; writing—original draft preparation, M.C.K., H.M.H., S.S., T.K.Z.E.-A., E.A.M. and H.O.E.; writing—review and editing, H.S.M., J.U.V., M.R.R. and S.V.; data curation, M.C.K., H.M.H., S.S., T.K.Z.E.-A., E.A.M. and H.O.E. All authors have read and agreed to the published version of the manuscript.

**Funding:** King Saud University (RSP-2021/118).

**Institutional Review Board Statement:** Not applicable.

**Informed Consent Statement:** Not applicable.

**Data Availability Statement:** All data are included in this publication.

**Acknowledgments:** The authors extend their appreciation to Researchers Supporting Project number (RSP-2021/118), King Saud University, Riyadh, Saudi Arabia, for their financial support for the publication of the present research manuscript. The authors would like to thank V. B. Nargund, Emeritus Professor and Former Head, Nanotechnology Laboratory, for helping in the characterisation of silver hydrogen peroxide. The authors extend our thanks to the Department of Plant Pathology, University of Agricultural Sciences, Dharwad, Karnataka, India, for facilitating needful requirements to conduct the experiment.

**Conflicts of Interest:** There were no conflicts of interest from the authors.

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
