# Peer review of "Colloidal Silver Hydrogen Peroxide: New Generation Molecule for Management of Phytopathogens"

_horticulturae, doi:10.3390/horticulturae7120573_

Round 1
Reviewer 1 Report
The current manuscript “Colloidal silver hydrogen peroxide: New generation molecule for plant disease management” states that silver hydrogen peroxide (SHP) possesses an anti-microbial activity against selected plant pathogenic fungi and bacteria under in vitro condition. The effective concentrations did not show any phytotoxicity on cotton plants. The current study opens the possibilities for further field validation of SHP against plant pathogens. However, I have some concerns regarding the presentation of the data.
- Hydrogen peroxide is a form of reactive oxygen species and can also act as signaling molecule for inducing host defense mechanisms. This could be self-toxic if not eliminated from hosts effectively. Addition of sliver could increase the host surface adhesion. Though it is not a novel idea, it is still a significant study in terms of practical application of the molecule. The discussion part of the study completely lacks possible mechanisms of anti-microbial property of SHP. Moreover, what are some of the shortcomings of the current study? Would SHP also affect biocontrol agent such as Trichoderma spp.?
- Why did the authors not test the same concentrations of SHP against bacteria and fungi?
- Legends of the figures are not self-explanatory. It does not include the number of technical and biological replicates. None of the legends are written well. Please provide more detailed legends including little methodology and replications etc.
- How many replications were there for phytotoxicity study? There is no statistical data provided. Please provide a data in a form of graph or table with statistical analysis. The pictures alone don’t provide any clear indication.
- Please go through the entire manuscript. Thera are lots of spaces missing between words.
Author Response
Manuscript ID: Horticulturae-1470721
Response to Reviewer 1 Comments
Point 1: Hydrogen peroxide is a form of reactive oxygen species and can also act as signaling molecule for inducing host defense mechanisms. This could be self-toxic if not eliminated from hosts effectively. Addition of sliver could increase the host surface adhesion. Though it is not a novel idea, it is still a significant study in terms of practical application of the molecule. The discussion part of the study completely lacks possible mechanisms of anti-microbial property of SHP. Moreover, what are some of the shortcomings of the current study? Would SHP also affect biocontrol agent such as Trichoderma spp.?
Response 1: Lower effective concentrations (100-2000 ppm) of silver hydrogen peroxide may not affect the growth of fungi like Trichoderma spp.. The aim of the study is to identify the effective doses of silver hydroxide peroxide against plant pathogenic fungi and bacteria. Because the application of silver hydrogen peroxide is scanty in agriculture as compared to medical science so the effort was made to generate preliminary data such as efficacy and phytotoxicity.
Point 2: Why did the authors not test the same concentrations of SHP against bacteria and fungi?
Response 2: The authors evaluated different concentrations (1000, 2000, 3000, and 4000 ppm) of SHP against fungal pathogens, and results were presented in line-261-262. Compared to bacteria, fungal pathogens require a higher concentration of chemicals due to their multi-cellular nature.
Point 3: Legends of the figures are not self-explanatory. It does not include the number of technical and biological replicates. None of the legends are written well. Please provide more detailed legends including little methodology and replications etc.
Response 3: Each treatment was replicated thrice and experiment was repeated twice. Legends were changed as per the suggestion.
Point 4: How many replications were there for phytotoxicity study? There is no statistical data provided. Please provide a data in a form of graph or table with statistical analysis. The pictures alone don’t provide any clear indication.
Response 4: In each treatment/concentration, three plants were used for phytotoxic study, and three leaves were used for the spotting of silver hydrogen peroxide. In phytotoxicity studies, the level of phytotoxicity will be recorded by using a visual phytotoxicity scale (0-10), and reference was added in the materials and method part.
Point 5: Please go through the entire manuscript. There are lots of spaces missing between words.
Response 5: Typographical errors are corrected in the manuscript.
Thank you for your valuable suggestion for the improvement of the manuscript.
Reviewer 2 Report
The paper title "Colloidal silver hydrogen peroxide: New generation molecule for plant disease management" by Mahesha, H. S. et al. submitted to Horticulturae reported the antimicrobial activity of silver hydrogen peroxide (SHP) against selected plant pathogenic fungi and bacteria under in vitro condition. Indeed, it is meaningful to do such a study and it would be even better to apply this technology in practice. However, at present stage, I found there are still many improvements that could be done by the authors especially for the manuscript writing and experiment set-ups.
- I am not a native English speaker; however, I did find many errors in this manuscript. I would suggest authors pay more attentions to the English polishing.
- I would suggest the author to reconsider the title carefully. Although it is quite attractive, it might not be precise enough considering the experimental design.
- Abstract: This section should be more concise. I don’t think it is necessary to mention the specific crops as background here (Line 19). What do authors mean ‘save time and production cost’? I didn’t see the reasonable result respect to this statement in the manuscript.
- Did authors test if the collected microbes (they call pathogens) are phytopathogenic based on Koch’s rule? Authors used large paragraphs to describe how to isolate the pathogens (Lines 94-121). However, I would suggest authors shorten this part. Instead, there should be many literatures describing classical isolation and they just cite these references or mention what kind of changes they used. Indeed, the antimicrobial test with silver hydrogen peroxide should be the main experiment in this study.
- Since there are many soil-borne pathogens collected, why not test the disease suppression of SHP in bioassays with plants?
- Lines 251-252 and 268: the meaning of significance should be indicated in the captions.
- Lines 254, 256, 258, 271: at least mention the scales of petri dishes here.
- Conclusions: This section should be reorganized since it reads not like a conclusion for this study.
- The authors should be more careful with the use of references. For example, it’s difficult to for me to follow the third reference.
Other comments:
Authors used the character ‘@’ in many places representing ‘at’. Is it ok with requirement of the journal?
Lines 22: ... was observed ...
Line 49: has -> have.
Lines 56-61: references needed.
Line 67: ‘is quickly decomposes’ ?
Lines 73-74: references needed.
Lines 76-77: references needed.
Line 80: add space (there are many errors in space missing throughout the manuscript)
Line 88: a pH ?
Author Response
Manuscript ID: Horticulturae-1470721
Response to Reviewer 2 Comments
Point 1: I am not a native English speaker; however, I did find many errors in this manuscript. I would suggest authors pay more attentions to the English polishing.
Response 1: English editing was done as for the suggestion.
Point 2: I would suggest the author to reconsider the title carefully. Although it is quite attractive, it might not be precise enough considering the experimental design.
Response 2: In the present study, we used both fungi and bacterial plant pathogens, which include both soil-borne as well as foliar pathogens. All these pathogens are having a wider host range. Hence authors decided to give the present title. But we are also comfortable with the following title “Antifungal and antibacterial activity of colloidal silver hydrogen peroxide on phytopathogenic fungi and bacteria”. The authors need the reviewer’s suggestion on this title.
Point 3: Abstract: This section should be more concise. I don’t think it is necessary to mention the specific crops as background here (Line 19). What do authors mean ‘save time and production cost’? I didn’t see the reasonable result respect to this statement in the manuscript.
Response 3: The crop background was removed in line 19. The authors thought that SHP showed both antifungal and antibacterial activity so that fungal and bacterial infections may be cured by a single application.
Point 4: Did authors test if the collected microbes (they call pathogens) are phytopathogenic based on Koch’s rule? Authors used large paragraphs to describe how to isolate the pathogens (Lines 94-121). However, I would suggest authors shorten this part. Instead, there should be many literatures describing classical isolation and they just cite these references or mention what kind of changes they used. Indeed, the antimicrobial test with silver hydrogen peroxide should be the main experiment in this study.
Response 4: The test organisms were isolated and their pathogenicity was proved with their respective hosts by satisfying Koch’s postulates. The isolation and characterization portion was reduced as per the suggestions.
Point 5: Since there are many soil-borne pathogens collected, why not test the disease suppression of SHP in bioassays with plants?
Response 5: The authors made effort to generate basic data on efficacy against different pathogenic microorganisms, phytotoxicity and particle size, etc related to silver hydrogen peroxide. The author’s next step is to take large-scale demonstrations in different crops under field conditions. So authors feel that to conduct large-scale demonstrations under field conditions, first-hand basic information is necessary. The information is lacking with respect to the use of SHP in the management of plant pathogens.
Point 6: Lines 251-252 and 268: the meaning of significance should be indicated in the captions.
Response 6: Figure caption was changed.
Point 7: Lines 254, 256, 258, 271: at least mention the scales of petri dishes here.
Response 7: Figure caption was changed as per the reviewer suggestion.
Point 8: Conclusions: This section should be reorganized since it reads not like a conclusion for this study.
Response 8: Conclusions part is reorganized
Point 9: The authors should be more careful with the use of references. For example, it’s difficult to for me to follow the third reference.
Response 9: The reference no. 3 was replaced with Kumar, S. 2014.
Others: Typographical errors were corrected and incorporated in the revised manuscript.
Thank you for your valuable suggestion for the improvement of the manuscript.
Round 2
Reviewer 1 Report
The manuscript has been improved to some extent. The current study lays a foundation for future research projects on the development of novel pathogen management strategies. The authors have NOT addressed the concerns completely. Still, I can’t find the discussion on the possible mechanisms of anti-microbial property of SHP. Moreover, the authors have not provided a figure of the data of the phytotoxicity assay. The leaves pictures cannot compensate for the real data.
Author Response
Manuscript ID: Horticulturae-1470721
Response to Reviewer 1 Comments
Comment 1: The manuscript has been improved to some extent. The current study lays a foundation for future research projects on the development of novel pathogen management strategies. The authors have NOT addressed the concerns completely. Still, I can’t find the discussion on the possible mechanisms of anti-microbial property of SHP. Moreover, the authors have not provided a figure of the data of the phytotoxicity assay. The leaves pictures cannot compensate for the real data.
Response 1: Mechanism of antimicrobial property of SHP was added (Line 309-312). Phytotoxicity data of SHP was added in the Table 4.
Reviewer 2 Report
In the revised version, authors have made many improvements for this manuscript. With respect to the title, if authors prefer the present one, they should at least use phytopathogen management instead of plant disease management, since they did not do bioassays here.
Line 31: why not keeping the abbr. for silver hydrogen peroxide?
Line 85: no superscript for H
Author Response
Manuscript ID: Horticulturae-1470721
Response to Reviewer 2 Comments
Comment 1: In the revised version, authors have made many improvements for this manuscript. With respect to the title, if authors prefer the present one, they should at least use phytopathogen management instead of plant disease management, since they did not do bioassays here.
Response 1: The article title was changed as per the suggestion and the final title is “Colloidal silver hydrogen peroxide: New generation molecule for management of phytopathogens”
Point 2: Line 31: why not keeping the abbr. for silver hydrogen peroxide?
Line 85: no superscript for H
Response 2: Typographical errors were corrected and incorporated in the revised manuscript.